# A single synonymous nucleotide change impacts the male-killing phenotype of prophage WO gene *wmk*

Jessamyn I Perlmutter[1,2,3]*, Jane E Meyers[1,3], Seth R Bordenstein[1,3,4,5]*

[1]Department of Biological Sciences, Vanderbilt University, Nashville, United States; [2]Department of Molecular Biosciences, University of Kansas, Lawrence, United States; [3]Vanderbilt Microbiome Innovation Center, Vanderbilt University, Nashville, United States; [4]Department of Pathology, Microbiology, and Immunology, Vanderbilt University, Nashville, United States; [5]Vanderbilt Institute for Infection, Immunology, and Inflammation, Vanderbilt University, Nashville, United States

**Abstract** *Wolbachia* are the most widespread bacterial endosymbionts in animals. Within arthropods, these maternally transmitted bacteria can selfishly hijack host reproductive processes to increase the relative fitness of their transmitting females. One such form of reproductive parasitism called male killing, or the selective killing of infected males, is recapitulated to degrees by transgenic expression of the prophage *WO-mediated killing* (*wmk*) gene. Here, we characterize the genotype-phenotype landscape of *wmk*-induced male killing in *D. melanogaster* using transgenic expression. While phylogenetically distant *wmk* homologs induce no sex-ratio bias, closely-related homologs exhibit complex phenotypes spanning no death, male death, or death of all hosts. We demonstrate that alternative start codons, synonymous codons, and notably a single synonymous nucleotide in *wmk* can ablate killing. These findings reveal previously unrecognized features of transgenic *wmk*-induced killing and establish new hypotheses for the impacts of post-transcriptional processes in male killing variation. We conclude that synonymous sequence changes are not necessarily silent in nested endosymbiotic interactions with life-or-death consequences.

*For correspondence:
jessamyn.perlmutter@ku.edu (JIP);
s.bordenstein@vanderbilt.edu (SRB)

## Editor's evaluation

This study identified the genetic mechanisms underlying sex-ratio distortion through male-killing in *Drosophila melanogaster* flies infected with the endosymbiont Wolbachia. The endosymbiont carries the prophage WO, which is the center of interest in this study. The key result of this study is that a synonymous mutation in a prophage gene can explain the differences between sex-ratio distorting and not distorting symbionts. The finding, that a synonymous SNP plays a key role is not entirely novel in biology, but there are only few examples known of this type of genotype–phenotype association.

## Introduction

*Wolbachia* are maternally transmitted, obligate intracellular bacteria primarily residing in the cells of germline tissues in many arthropod species worldwide (*Hurst and Frost, 2015*; *Taylor et al., 2018*.) To facilitate their spread through the host matriline, these bacteria hijack sex ratios, sex determination systems, and embryonic viability to cause various reproductive parasitism phenotypes (*Yen and Barr, 1971*; *Hurst et al., 1999*; *Bouchon et al., 1998*; *Hunter, 1999*). One such phenotype is male killing, whereby sons of infected females are selectively killed (*Hurst et al., 1999*; *Hurst and Jiggins,*

*2000*; *Charlat et al., 2005*). Male killing selfishly drives the spread of the bacteria when, for instance, brothers and sisters compete for limited resources, and male lethality yields females (the transmitters) a relative fitness benefit due to reduced competition with fewer siblings (*Jaenike et al., 2003*; *Unckless and Jaenike, 2012*; *Skinner, 1985*; *Hurst, 1997*). This form of reproductive parasitism can have profound impacts on host behavior and evolution, including a sex role reversal (*Jiggins et al., 2000*), host nuclear genome changes to resist the phenotype (*Majerus and Majerus, 2010*; *Hornett et al., 2006*), and potentially host population extinction (*Groenenboom and Hogeweg, 2002*). Population modelling also suggests male killing may be deployed as a control method to crash the population size of arthropod pests and disease vectors (*Berec et al., 2016*).

Many lines of evidence suggest a complex relationship between male-killing genotype and phenotype. For example, male killing can be suppressed by host background (*Majerus and Majerus, 2010*; *Hornett et al., 2006*; *Jaenike, 2007*; *Mitsuhashi et al., 2011*), where male hosts exhibit resistance to the phenotype even in the presence of *Wolbachia*. In addition, *Wolbachia* that do not cause male killing in one host species can cause male killing in a second host species upon introgression or transfer to a naïve host (*Jaenike, 2007*; *Fujii et al., 2001*). Furthermore, some strains induce death at different host developmental stages (embryo vs. larvae) or across different sex determination systems (ZW lepidopterans, XY dipterans, XO arachnids), and the number of surviving males may vary widely (*Riparbelli et al., 2012*; *Sasaki et al., 2002*; *Zeh et al., 2005*). Such findings specify that the expression of male killing, and particularly the genotype-phenotype relationship in these symbioses, is more complex than simply the presence or absence of a male-killing gene.

We recently identified a male-killing candidate gene, *wmk* (*WO-mediated killing*), from prophage WO in the *w*Mel *Wolbachia* strain of *Drosophila melanogaster* based on comparative genomic, transgenic, and cytological approaches (*Bordenstein and Bordenstein, 2016*; *Perlmutter et al., 2019*). It is a putative transcription factor with two predicted helix-turn-helix (HTH) DNA-binding domains, and transgenic expression in *D. melanogaster* embryos recapitulates many cytological and molecular aspects of male killing including accumulation of DNA damage in male embryos that overlaps with sites of dosage compensation activity (*Riparbelli et al., 2012*; *Perlmutter et al., 2019*; *Harumoto et al., 2018*). The *wmk* gene and two cytoplasmic incompatibility factor (*cif*) genes that underlie cytoplasmic incompatibility (a parasitism phenotype whereby offspring die in crosses between infected males and uninfected females) occur nearby specifically in the eukaryotic association module (EAM) of prophage WO (*Bordenstein and Bordenstein, 2016*; *Perlmutter et al., 2019*; *LePage et al., 2017*), which refers to the phage genome that is inserted into the bacterial chromosome. The EAM is common in WO phages across several *Wolbachia* strains (*Masui et al., 2001*; *Wu et al., 2004*) and is rich in genes that are homologous to eukaryotic genes or annotated with eukaryotic functions (*Bordenstein and Bordenstein, 2016*). As such, the expression of reproductive parasitism genes from the EAM and tripartite interactions between phage WO, *Wolbachia*, and eukaryotic hosts are central to *Wolbachia*'s ability to interact with and modify host reproduction.

The discovery of *wmk* has now enabled investigation of the impacts of genetic variation on the transgenic male-killing phenotype. Although the role of *wmk* in male killing has not yet been assessed in nature, evidence based on native *wmk* genotypes and transgenic *wmk* expression in flies suggest refined intricacy to the interactions between genotype and phenotype. For example, the native *Wolbachia wmk* homologs in three closely-related strains (*w*Mel of *D. melanogaster*, *w*Rec of *D. recens*, and *w*Suzi of *D. suzukii*) have nearly identical nucleotide sequences that vary by one to three SNPs (*Perlmutter et al., 2019*), and yet these strains induce different forms of reproductive parasitism. *w*Mel and *w*Rec induce cytoplasmic incompatibility, *w*Rec also causes male killing in some strains of a sister species, and *w*Suzi is not known to induce any reproductive phenotype (*Jaenike, 2007*; *Hamm et al., 2014*; *Hoffmann, 1988*). Previous transgenic testing also demonstrated that addition of nine additional amino acids to the N-terminal region of the Wmk protein at the site of a putative alternative start codon ablates the phenotype (*Perlmutter et al., 2020*). Therefore, the Wmk N-terminus is particularly sensitive to genetic alterations, which is notable since small modifications such as protein tags in this location are often common additions to proteins that do not typically interfere with function. Furthermore, *D. melanogaster* does not harbor *Wolbachia* known to cause male killing, although transgenic *wmk* expression recapitulates the phenotype in this host (*Perlmutter et al., 2019*), and the *w*Mel strain is most closely-related to the *w*Rec male-killing strain (*Metcalf et al., 2014*). In addition, some *D. melanogaster* contain

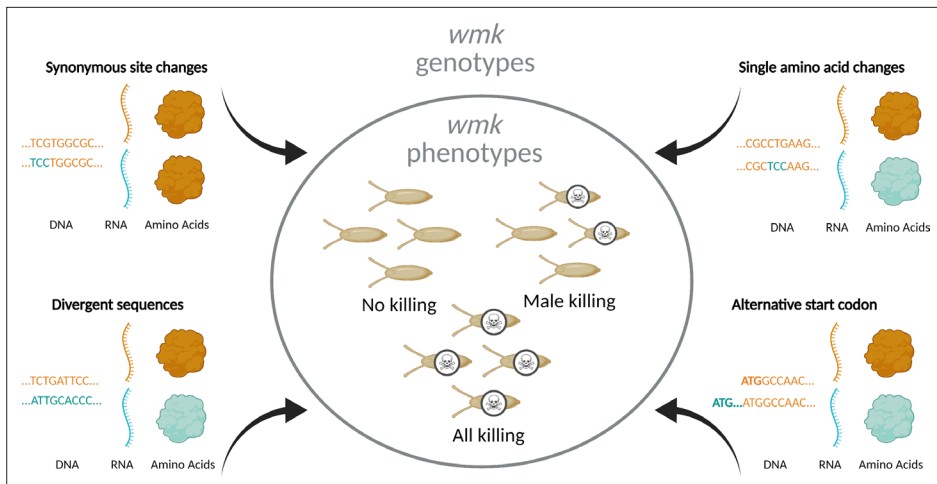

**Figure 1.** Overview of experimental design. To investigate the genotype-phenotype landscape, we transgenically expressed *wmk* homologs with varying degrees of genetic changes. These sequences are codon-optimized based on different codon biases due to different tRNA abundances in the divergent bacterial source and eukaryotic destination species (*Plotkin and Kudla, 2011*; *Gustafsson et al., 2004*). Transgenic *wmk* in *Drosophila melanogaster* embryos results in three different phenotypes: no killing, male killing, and killing of males and females. Compared to *w*Mel *wmk*, these transgenes were either divergent homologs from other *Wolbachia* strains, a homolog with a single amino acid change, homologs with an additional nine codons at the 5' ends of the genes starting at an alternative upstream start codon, or variants with a single synonymous codon or nucleotide difference. These genotypes resulted in varying degrees of RNA sequence- and amino acid-level changes. Created with BioRender.com.

The online version of this article includes the following figure supplement(s) for figure 1:

**Figure supplement 1.** Homologs of *wmk* tested in this study include variation in native gene and transgene sequence identity as well as host species.

male-killing *Spiroplasma* symbionts whose plasmid gene *Spaid* can also transgenically recapitulate male killing (*Harumoto and Lemaitre, 2018*). In summary, the evidence for *Wolbachia* and *wmk* transgenic male killing suggests an intricate and multifaceted genotype-phenotype landscape.

Building on this background, several key questions emerge: Do closely-related or distantly-related homologs induce male killing? How sensitive is the *wmk* phenotype to small or large genetic changes? And how adapted are *wmk* homologs to their arthropod hosts? Here, we evaluate codon-optimized *wmk* homologs that span a spectrum of genetic divergence (*Figure 1*), including homologs (i) from distantly-related hosts, (ii) with putative upstream alternative start codons, (iii) with a single amino acid change, or (iv) with one or more synonymous codon changes, all in the *D. melanogaster* host. In this way, we aimed to investigate how a variety of genetic alterations to *wmk*, from a single synonymous nucleotide to the gene level, affect the phenotype. In particular, we hypothesized that divergent strains would not induce the phenotype due to co-adaptation with a distantly-related host. We also anticipated that the alternative start codons would inhibit function based on previous results (*Perlmutter et al., 2020*). In contrast, we hypothesized that neither a single amino acid change nor synonymous codon changes would alter the male-killing phenotype. We report that while distant homologs do not cause a male-killing phenotype in this host, single amino acid and synonymous nucleotide changes remarkably do alter the phenotype. Thus, *wmk* male killing is sensitive to the full spectrum of genetic alterations at a fine-scale level not previously recognized. Notably, synonymous sequence changes and post-transcriptional processes appear to play a role in controlling the genotype-phenotype relationships that underpin *wmk* male killing.

## Results
### Closely-related homologs of wmk
We tested homologs from strains related to *w*Mel that occur in several groups of species in the *Drosophila* genus, along with two from Order Lepidoptera (*Figure 1—figure supplement 1*). The

phylogenetic clades of both the native *Wolbachia* genes and their transgenes are similar to those of the host species. Roughly divided, the transgenes group into two clusters that offer a range of genetic divergence for evaluating genotype-phenotype relationships (*Figure 1—figure supplement 1*) - those distantly related to the *w*Mel *wmk* transgene (less than 90 % codon-optimized nucleotide sequence similarity to transgenic *w*Mel *wmk*) and those closely related (greater than 90 % identity, from *w*Rec, *w*Suzi, and an HA-tagged *w*Mel homolog).

The closely-related *wmk* homologs to *w*Mel include those from *Wolbachia* strains (i) *w*Suzi of the fruit pest species *D. suzukii* that has no confirmed reproductive phenotype but notably occurs in populations with female-biased sex ratios (*Hamm et al., 2014*; *Drummond et al., 2019*) and (ii) *w*Rec of *D. recens* that kills males when introgressed into its sister species, *D. subquinaria* (*Jaenike, 2007*). The natural *w*Suzi *wmk* homolog has one synonymous single nucleotide polymorphism (SNP) compared to the *w*Mel *wmk* reference, thus yielding the same amino acid sequence. The natural *w*Rec *wmk* homolog has three SNPs, one of which is non-synonymous relative to *w*Mel *wmk*, located in the first HTH DNA-binding domain (*Figure 2A*). These transgenes were codon optimized for expression in *D. melanogaster*. We hypothesized that both transgenes would induce a biased sex ratio comparable to *wmk* when expressed in *D. melanogaster*. However, transgenic expression of both homologs unexpectedly resulted in death of all expressing flies, both male and female (*Figure 2B*). In addition, we simultaneously tested a transgene of *w*Mel *wmk* with an internal 3 X HA tag epitope in the linker region between the two HTH DNA-binding domains. This transgene, HA-*wmk*, exhibited a sex ratio bias comparable to *w*Mel *wmk*, as expected (*Figure 2B*). Therefore, two transgenes of *w*Mel *wmk* (one with a tag) resulted in the biased sex ratio previously reported (*Perlmutter et al., 2019*), while two closely-related transgenes from strains that at least associate with female-biased host sex ratios yielded an all-killing phenotype.

## Gene expression similarities

To assess if variation in gene expression underpins the phenotypic variation, we measured the transcript levels of the transgenes in embryos 4–5 h AED (after egg deposition) when *wmk* kills males (*Perlmutter et al., 2019*). Gene expression levels are not significantly different across *w*Suzi, *w*Mel, *w*Rec, or HA tag *w*Mel *wmk* transgenes, indicating that transcript levels do not account for the phenotypic differences (*Figure 2C*). An alternative hypothesis is that transgenic expression of the *w*Suzi and *w*Rec homologs impacts native expression of a host gene in *D. melanogaster* that causes male and female lethality. For instance, transgenic expression of the DNA-binding dosage compensation gene, *male-specific lethal 2* (*msl-2*), can induce total lethality with male-killing *Spiroplasma* (*Cheng et al., 2016*). Based on this, we quantified *msl-2* transcript levels in embryos expressing *w*Mel *wmk* (sex ratio bias), *w*Suzi *wmk* (all expressing hosts die), and *w*Bif *wmk* (no killing or sex ratio bias) and found that *msl-2* levels were comparable across all genotypes and phenotypes (*Figure 2D*).

## RNA structural model variation

We next considered mRNA secondary structural differences of the transgene transcripts as a factor explaining the observed phenotypic variation. Modeling RNA structures showed they were substantially different, even in the case of few to no amino acid level changes from one sequence to another. Notably, some RNA structural features grouped by phenotype, such as the location of the start codon (*Figure 2E*). Two transgenic transcripts, *w*Mel *wmk* and HA-*wmk*, cause a sex ratio bias and have start codons in the middle of the structure, while the other transgenes, *w*Suzi *wmk* and *w*Rec *wmk*, that kill all flies have start codons on outer loops or hairpins. Although caution is warranted with this predicted structural analysis, mRNA secondary structure could explain some phenotypic outcomes of closely-related *wmk* homologs. Native *w*Mel *wmk* also has a start codon on an outer loop but does not induce an 'all killing' phenotype like the *w*Suzi and *w*Rec *wmk* transgenes, which may be due to lower expression of native genes (*Perlmutter et al., 2019*) and/or other structural differences.

## *Distantly-related homologs of* wmk

To determine if *wmk* homologs from distantly-related strains induce a biased sex ratio in *D. melanogaster*, we transgenically expressed four codon-optimized homologs from known male-killing strains of *Wolbachia*: the *w*Bol1b strain from *Hypolimnas bolina* butterflies (*Dyson et al., 2002*), the *w*Bif strain from *Drosophila bifasciata* flies (*Riparbelli et al., 2012*), the *w*Caub strain from *Cadra cautella*

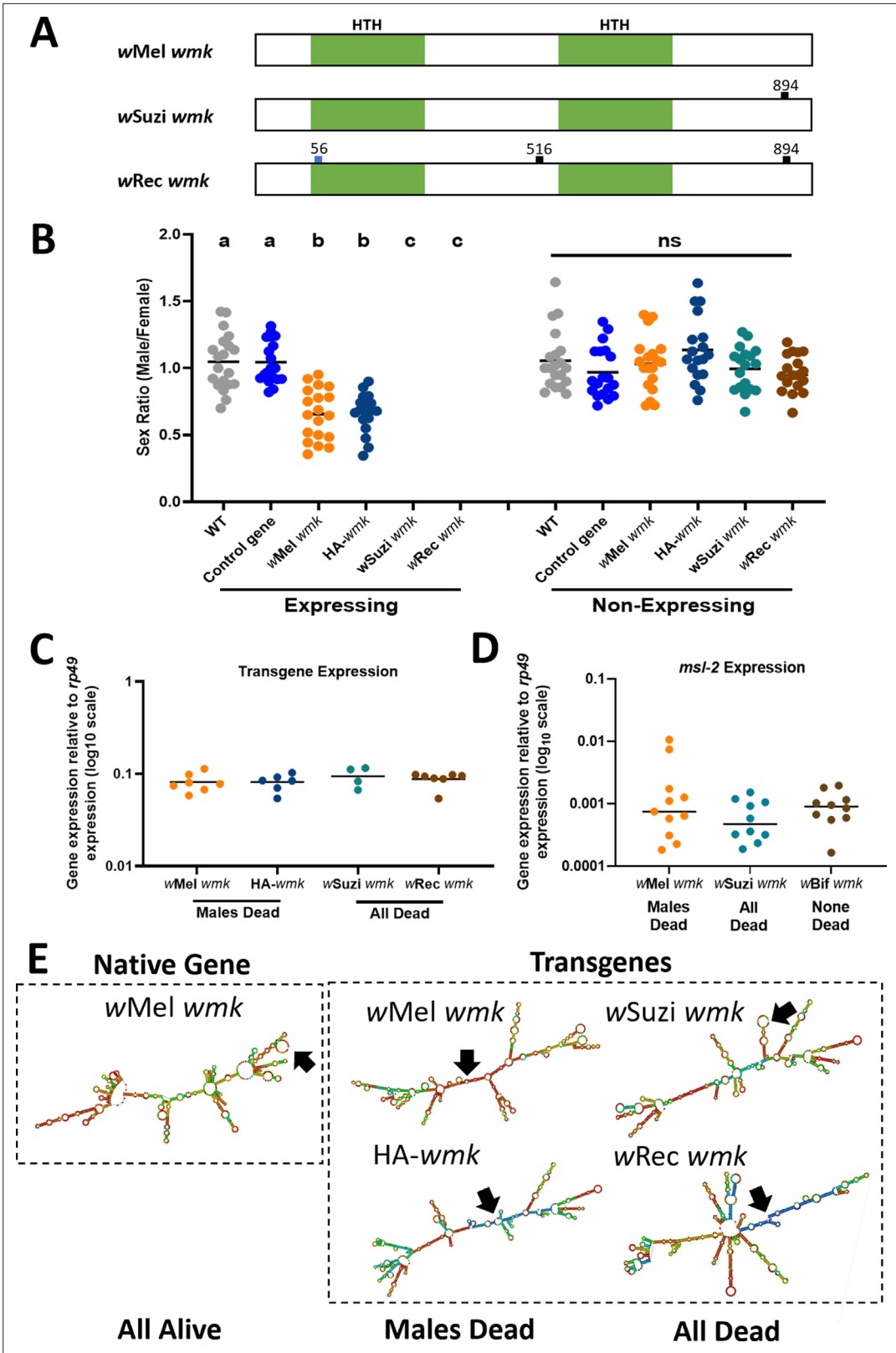

**Figure 2.** Transgenic expression of closelyrelated *wmk* homologs causes male-killing and all-killing phenotypes in *D. melanogaster*. (**A**) Schematic of *w*Mel, *w*Suzi, and *w*Rec *wmk* native nucleotide sequences. The blue tick mark indicates a non-synonymous nucleotide difference. Black tick marks indicate synonymous nucleotide changes. Numbers indicate nucleotide position across the entire 912 nucleotide sequence. (**B**) Sex ratios of adult flies are

*Figure 2 continued*

shown for expressing (*Act5c*-Gal4) and non-expressing (CyO) embryonic offspring. Each sample point represents the adult offspring (N = 50–157, mean 86) produced by a replicate family of ten mothers and two fathers, with expressing and non-expressing flies of a given genotype being siblings. Bars represent the mean sex ratio. Statistics are based on a Kruskal-Wallis, one-way ANOVA followed by Dunn's correction across either expressing or non-expressing flies. *w*Rec and *w*Suzi *wmk* have no points in the expressing category due to death of most or all males and females. HA-*wmk* contains a 3 X HA tag in the linker region between the two helix-turn-helix domains. This experiment was performed twice. Data and statistical outputs are available in *Figure 2—source data 1* and *Figure 2—source data 2*, respectively. (**C**) Gene expression in embryos 4–5 h AED of each indicated *wmk* transgene from (**B**), relative to *Drosophila* housekeeping gene, *rp49*. There is no significant difference in expression based on a Kruskal-Wallis one-way ANOVA followed by Dunn's correction. Data and statistical outputs are available in *Figure 2—source data 3* and *Figure 2—source data 4*, respectively. (**D**) Gene expression in embryos 4–5 h AED of the host *msl-2* dosage compensation gene relative to *rp49* under simultaneous expression of the indicated transgene. There is no significant difference in expression based on a Kruskal-Wallis one-way ANOVA followed by Dunn's correction. Data and statistical outputs are available in *Figure 2—source data 5* and *Figure 2—source data 6*, respectively. (**E**) Predicted RNA secondary structures of native *w*Mel *wmk* and several transgene strains. Black arrows point to the location of the start codon within each structure.

The online version of this article includes the following source data and figure supplement(s) for figure 2:

**Source data 1.** Data for sex ratios of closely-related homologs corresponding to *Figure 2B*.

**Source data 2.** Statistical output of Kruskal-Wallis test corresponding to sex ratios of closelyrelated homologs in *Figure 2B*.

**Source data 3.** Data for qPCR of closely-related transgenes corresponding to *Figure 2C*.

**Source data 4.** Statistical output of Kruskal-Wallis test corresponding to qPCR for transgene expression in *Figure 2C*.

**Source data 5.** Data for qPCR of msl-2 expression with transgene expression corresponding to *Figure 2D*.

**Source data 6.** Statistical output of Kruskal-Wallis test corresponding to qPCR for msl-2 expression in *Figure 2D*.

**Figure supplement 1.** *w*Rec and *w*Suzi transgenes expressed with an alternative start codon lose their transgenic phenotypes.

**Figure supplement 1—source data 1.** Data for sex ratios of 5' alternative start codon transgene expression corresponding to *Figure 2—figure supplement 1*.

**Figure supplement 1—source data 2.** Statistical output of Kruskal-Wallis test corresponding to sex ratios of 5' alternative start codon transgene expression in *Figure 2—figure supplement 1*.

moths (*Sasaki et al., 2002*), and the *w*Inn strain from *D. innubila* flies (*Dyer and Jaenike, 2004*) (same gene sequence as the *w*Bor male-killing strain from *D. borealis* flies *Carson, 1956*; *Figure 1—figure supplement 1*). While transgene expression of *w*Mel *wmk* induces a biased sex ratio (~one third of expressing males die), none of the more distantlyrelated *w*Bol1b, *w*Bif, *w*Caub, or *w*Inn/*w*Bor *wmk* homologs yield a biased sex ratio, demonstrating that they do not recapitulate male killing when transgenically expressed in *D. melanogaster* under the conditions tested (*Figure 3*).

## Alternative start codon variation and male killing

Relevant to the studies here, we previously provided evidence that some strains contain alternative start codons upstream of the annotated start for *wmk,* and these upstream regions are expressed in the *w*Mel strain (*Perlmutter et al., 2020*). When *w*Mel *wmk* was transgenically expressed with the most likely upstream start codon, the phenotype was lost, and no biased sex ratio resulted. We also showed that some non-male-killing strains tended to have more of the alternative start codons (*Perlmutter et al., 2020*). To determine if *w*Rec and *w*Suzi *wmk* transgene phenotypes are similarly sensitive to transcript changes, we expressed them with upstream codons that are native to each of their genetic sequences. As previously observed with other homologs, they lost their killing phenotype with only nine amino acids added to the 5' end of the gene, despite being smaller than many commonly used protein tags (*Figure 2—figure supplement 1*). All expressing flies survived with no sex ratio bias. Returning to the RNA structure models, we find that simply adding the corresponding nucleotides at the 5' end of each homolog resulted in several predicted differences in RNA secondary structure for each transgene compared to the structures without the additional 5' nucleotides (*Figure 2—figure supplement 1*). This includes additional loops and different predicted placement of the start codon.

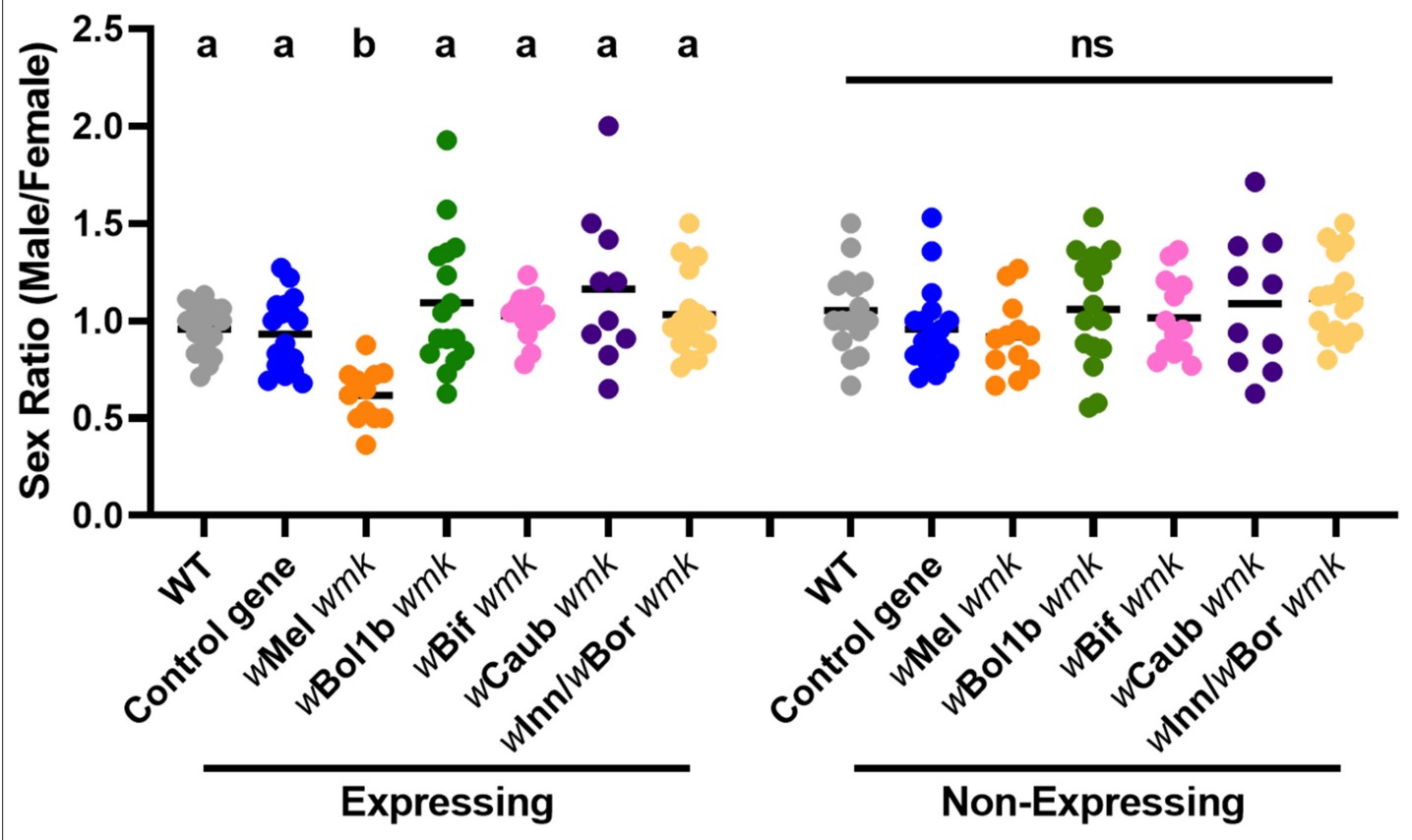

**Figure 3.** Divergent homologs of *wmk* from male-killing strains do not induce a biased sex ratio in *D. melanogaster*. Sex ratios of adult flies are shown from either expressing (*Act5c*-Gal4) or non-expressing (CyO) offspring. WT refers to the background insertion line and Control gene refers to the WD0034 control transgene that induces no sex ratio bias. Each sample point represents the adult offspring (N = 50–132, mean 69) produced by a replicate family of ten mothers and two fathers, with expressing and non-expressing flies of a given genotype being siblings. Bars represent the mean sex ratio. Statistics are based on a Kruskal-Wallis one-way ANOVA followed by Dunn's correction across either expressing or non-expressing flies. This experiment was performed twice. Data and statistical outputs are available in *Figure 3—source data 1* and *Figure 3—source data 2*, respectively.

The online version of this article includes the following figure supplement(s) for figure 3:

**Source data 1.** Data for sex ratios of distantly-related homologs in *Figure 3*.

**Source data 2.** Statistical output of Kruskal-Wallis test corresponding to sex ratios of divergent homologs in *Figure 3*.

### Silent site variation and male killing

Finally, to identify particular nucleotides that may account for phenotypic variation among the homologs, we aligned the sequences of the four transgenes in *Figure 2* and investigated codons that clustered by phenotype (sex ratio bias for *w*Mel *wmk* and HA-*wmk*, or all killing for *w*Rec and *w*Suzi *wmk*). Across the length of the genes (and excluding the HA tag), there were only two codon differences: one at the sixteenth amino acid position and another near the end of the gene. As previous work demonstrated that changes at the 5' end of this gene affect phenotype (*Perlmutter et al., 2020*) and since approximately the first 10 codons in model prokaryotic genes are known to substantially affect mRNA structure and resulting translation rate (*Bentele et al., 2013*), we focused on the earlier codon at site 16. This codon, which codes for Serine in all homologs, segregates among sequences by phenotype. The HA-*wmk* and *w*Mel *wmk* transgenes, which recapitulate male killing, have a TCG codon, while the all-killing transgenes *w*Suzi and *w*Rec *wmk* have TCC and AGC, respectively (*Figure 4A*).

To functionally test if the silent changes in this Serine codon accounted for the phenotype differences, we generated three new transgenes (*Figure 4B*): (i) *w*Mel *wmk* control with no changes compared to the previously tested transgene (TCG codon, labeled '*w*Mel *wmk* (new)'), (ii) *w*Mel *wmk* with three nucleotide changes in the codon that reconstitutes the AGC present in the *w*Rec transgene line (labeled '*w*Rec codon'), and (iii) *w*Mel *wmk* with one nucleotide change in the codon that

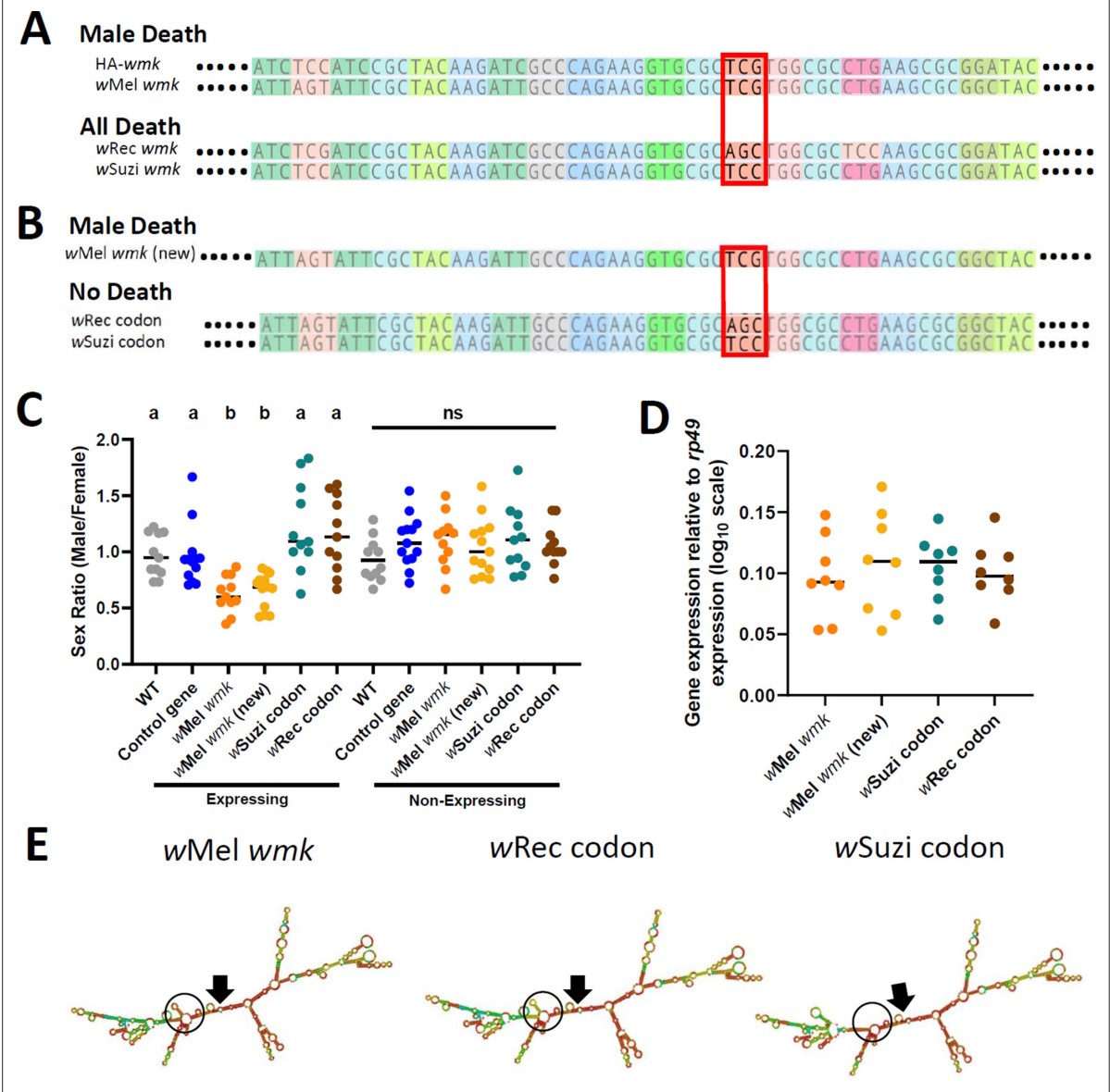

**Figure 4.** Synonymous nucleotide changes in the 16th codon position of *wmk* alters resulting phenotype. (**A**) Sequence alignment of transgenic *wmk* homologs. The codon farthest on the left is the fourth codon in the sequence, and the highlighted codon is the 16th, with the farthest right representing the 23rd codon, and ellipses indicating codons continuing on either side. The red box outlines where the genotypes cluster by phenotype. The 'HA-*wmk*' and '*w*Mel *wmk*' genotypes share the same codon in this position, and both induce male-specific death. The '*w*Rec *wmk*' and '*w*Suzi *wmk*' genotypes both exhibit different codons from the previous two and exhibit an all-killing phenotype. Colors correlate with amino acid identity. (**B**) Sequence alignment of transgenes with either the *w*Mel *wmk* sequence made anew (*w*Mel *wmk* new), or with the 16th codon (red box) replaced with the synonymous codons from the *w*Rec and *w*Suzi *wmk* transgenes. The colors and symbols reflect those in (**A**). (**C**) Sex ratios of adult transgenic flies are shown for expressing (*Act5c*-Gal4) and non-expressing (CyO) offspring that include the original transgene *w*Mel *wmk* strain used in previous figures, along with the newly created identical *w*Mel *wmk* (new) transgene and the additional transgenes with the single codon swapped out for the indicated codons noted in (**A**). *w*Suzi codon and *w*Rec codon refer to the strains that have the same sequence as the *w*Mel *wmk*, but with one or three silent sites changed in the single codon at the 16th amino acid position. Each sample point represents the adult offspring (N = 50–161, mean 73) produced by a replicate family of 10 mothers and two fathers, with expressing and non-expressing flies of a given genotype being siblings. Bars represent the mean sex ratio. Statistics are based on a Kruskal-Wallis one-way ANOVA followed by Dunn's correction across either expressing or non-expressing flies. This experiment was performed twice. Data and statistical outputs are available in *Figure 4—source data 1* and *Figure 4—source data 2*, respectively. (**D**) Gene expression in embryos 4–5 h AED denotes expression of each transgene relative to that of *rp49*. There is no significant difference in expression based on a Kruskal-Wallis one-way ANOVA followed by Dunn's correction. Data and statistical outputs are available in *Figure 4—source data 3* and *Figure 4—source data 4*, respectively. (**E**) Predicted RNA secondary structures are shown for the *w*Mel *wmk* transcript compared to both of the *w*Rec or *w*Suzi codon transgenes exhibiting slight structural differences. The structure for transgene *w*Mel *wmk* is included again from *Figure 3* for ease of

*Figure 4 continued on next page*

*Figure 4 continued*

comparison. Black arrows point to the location of the start codon within each structure. Black circles highlight a key area of difference between the structures, with a stem absent in the *w*Suzi codon strain, and different base pair match probabilities calculated for each as indicated by color (blue to red, low to high probability). Within the black circle, the *w*Suzi codon transgene structure is missing a predicted stem that the others have. The stem in the *w*Rec codon line, while present, has a weaker prediction as noted by the cooler colors, so there may be structural differences compared to the *w*Mel *wmk* model.

The online version of this article includes the following figure supplement(s) for figure 4:

**Source data 1.** Data for sex ratios from expression of transgenes with single codon changes corresponding to *Figure 4C*.

**Source data 2.** Data for qPCR from expression of transgenes with single codon changes corresponding to *Figure 4D*.

**Source data 3.** Data for qPCR from expression of transgenes with single codon changes corresponding to *Figure 4D*.

**Source data 4.** Statistical output of Kruskal-Wallis test corresponding to qPCR from expression of transgenes with single codon changes in *Figure 4D*.

reconstitutes the TCC present in the *w*Suzi transgene line (labeled '*w*Suzi codon'). When these three otherwise identical genes were transgenically expressed, the wMel *wmk* (new) transgene with no changes caused a biased sex ratio as expected; however, and remarkably, expression of transgenes with two different Serine codons ablated the phenotype and resulted in a non-biased sex ratio with normal numbers of expressing flies (*Figure 4C*). This ablation occurs even though transcript levels remain similar across all transgenes and despite sequencing confirmation of the single codon differences (*Figure 4D*). Thus, a minimum of one single synonymous site change in the 5' region was sufficient to alter the sex ratio phenotype. However, while the nucleotide changes in the codon changed the phenotype, they did not recapitulate the all-killing phenotypes of their corresponding homologs. The predicted RNA secondary structures from the transgenes with the single codon changes are similar to the original *w*Mel *wmk* transgene, but they differ in some aspects such as presence or absence of stems and loops and the probability score of the base pair match as indicated by color (scale of red to blue, warmer colors indicate high probability, cooler colors indicate low probability). (*Figure 4E*).

## Discussion

Linking mutations to function is crucial in resolving the evolutionary dynamics of adaptations, especially when the functions emerge in a nested system of tripartite phage-bacteria-animal interactions. In a simple case, the direct impact of genetic divergence is functional divergence with increasing numbers of mutations leading to increased functional divergence. Here, we investigated evolutionary and molecular hypotheses related to the genotype-phenotype relationships between the prophage *WO-mediated killing* (*wmk*) gene (*Perlmutter et al., 2019*; *Perlmutter et al., 2020*) and male killing in *D. melanogaster*. Sequences among *wmk* homologs in nature can differ across divergent hosts, leading to the hypothesis that highly divergent alleles are functionally fine-tuned. Using a variety of sequences, we report three key results: (i) mutations even at the synonymous codon and single nucleotide levels can alter the male-killing phenotype, (ii) phenotype, genotype, and RNA structural variation exhibit some correlation with each other, and (iii) distantly-related homologs do not induce a male-killing phenotype in transgenic *D. melanogaster*, while some closely-related homologs from *Drosophila* species/populations with female-biased sex ratios induce all-killing phenotypes. We discuss how these findings expand and support mechanistic models of *wmk* male killing, emphasize important implications for transgenic assays in endosymbiont studies, and relate these findings to aspects of male killing such as phenotype switching and host resistance.

The most surprising and unanticipated result was that transgene expression of highly similar homologs and even single synonymous site changes alter phenotype, and they make the difference between life and death for some males. We tested *wmk* homologs with high native sequence identity compared to native *w*Mel *wmk*: *w*Rec *wmk* from the mushroom-feeding *D. recens* (native sequence has two synonymous and one non-synonymous nucleotide differences) and *w*Suzi *wmk* from the fruit crop pest *D. suzukii* (native sequence has one synonymous nucleotide difference). Although we anticipated similar results to *w*Mel *wmk* expression, we found that transgenic expression of these genes killed all flies (*Figure 2*), even though the *w*Suzi *wmk* transgene produces an identical protein to transgenic *w*Mel Wmk. Of particular note, transgenic *w*Suzi *wmk* (all killing) and the original transgenic

*w*Mel *wmk* sequence (male killing) produce proteins of the exact same amino acid sequence, but they shared only a 91 % sequence identity in the codon-optimized transgene, largely due to an updated codon optimization algorithm. Codon optimization is the norm in transgenic symbiosis research under the common assumption that synonymous codons are functionally redundant. However, codon optimization algorithms often choose a codon based on factors including codon adaptation, mRNA folding, regulatory motifs, nucleotide bias, or codon correlations and biases (*Plotkin and Kudla, 2011*). With even a few different codons input into the algorithm as well as algorithm updates over the years, the tested transgenes had different nucleotide sequences.

Based on the aforementioned results, we next sought to assess the sensitivity of male killing to *wmk* transcriptional and post-transcriptional changes and asked if one synonymous codon or site change was sufficient to alter phenotype. We created three transgenic lines, each with different DNA sequences coding for the same Serine at position 16. With only these minor changes that encode an identical protein sequence to *w*Mel Wmk, the transgenic male-killing phenotype was ablated (*Figure 4*). Thus, one single codon and even a single nucleotide, remarkably determined male viability. Importantly, this key result implies at the molecular level that endosymbiont phenotypes of reproductive parasites may not simply be governed by DNA sequence alone.

A key question related to our findings is how synonymous changes cause vastly different phenotypes. It is often assumed in endosymbiont transgenic experiments that synonymous changes do not affect function, and codons for the same amino acid are functionally redundant. However, several decades of research have uncovered mounting evidence that the functional redundancy hypothesis is not always accurate (*Plotkin and Kudla, 2011*). Indeed, codon bias varies across species (hence the need for codon optimization) (*Sharp et al., 1997*; *Andersson and Kurland, 1990*), and there are several mechanisms through which single or few codons can influence transcription or translation. When there is a rare codon, there are fewer tRNAs available, and the translation rate may be correspondingly lowered (*Li et al., 2012*). Notably, there is a bias for rare codons in N-terminal regions of bacterial genes, which is likely due to their influence on mRNA structure (*Goodman et al., 2013*). Lower 5' RNA stability or weaker structure can also be correlated with ease of initiation and faster translation (*Gu et al., 2010*), and stem loop structures may affect translation rate in varying ways. Indeed, strong 5' RNA secondary structure may inhibit ribosomal initiation or translational efficiency (*de Smit and van Duin, 1990*; *Bettany et al., 1989*), and there can be selection against SNPs that alter RNA secondary structure (*Salari et al., 2013*). There is also evidence for selection across the domains of life on codon usage in the first 30–60 nucleotides of a gene, likely due to their impact on mRNA structure near the site of initiation (*Gu et al., 2010*). Additionally, early codons with low frequencies can be beneficial by slowing elongation, thus preventing ribosome collisions, or potentially helping to recruit chaperones to the emerging peptide for proper folding (*Tuller et al., 2010*; *Fredrick and Ibba, 2010*). In addition, some specific codons across the sequence are favored for their lower likelihood of mistranslation (*Qin et al., 2004*). Indeed, synonymous codon differences in genes can result, for instance, in altered gene expression (*Kudla et al., 2009*), lower organism fitness (*Agashe et al., 2013*), or disease (*Sauna and Kimchi-Sarfaty, 2011*).

Beyond codon sequence, gene expression levels across representatives of all phenotypes are similar (*Figure 2*), supporting post-transcriptional differences as the source of functional variation and refuting the functional redundancy hypothesis in this context. Examining the three synonymous transgenes more closely, we find that codon usage indeed correlates with phenotype. According to the codon usage table for *D. melanogaster* on the Genscript website, the original *w*Mel *wmk* codon at this position (TCG) has a frequency 16.7 per 1000 codons. In contrast, the *w*Suzi codon (TCC) has a frequency of 19.5/1000, and the *w*Rec codon (AGC) has a frequency of 20.5/1000 (*GenScript, 2020*). Therefore, there is a pattern where the more common codons are present in the all-killing transgenes, and at similar frequencies, while the strain expressing partial male killing has a codon at a lower frequency. Thus, some of the phenomena described in studies on synonymous codons and function likely underlie the phenotypic variation observed here.

Given that transgenic *wmk* expression with silent site changes results in different phenotypes, we used software that estimated RNA secondary structures for the tested transgenes. Indeed, we found that the predicted structures correlate somewhat with phenotype, so synonymous codons may change transcript structure or other post-transcriptional features of the transcript or protein (*Figure 2*). Other small changes including adding a 9-codon N-terminal sequence ablated the phenotype and reversed

it to no death, despite the additional sequence being smaller than many common gene tags that typically do not interfere with function. This small change also alters predicted RNA structure (*Figure 2—figure supplement 1*). We were unable to assess protein levels to determine translational differences due to lack of an antibody. However, the data demonstrates that the *wmk* transgene is functionally sensitive to some post-transcriptional changes, at least in the N-terminal region.

What, then, do these results illuminate about transgenic *wmk* function and the utility of transgenic research in light of the potential for marked influences of synonymous changes on phenotype? First, as discussed, the *wmk* transgenic phenotypes are likely sensitive to post-transcriptional processes. This has important implications for understanding *wmk* in a natural male-killing context as soon as feasible techniques are available, since heterologous expression and its reliance on codon optimization may obscure our understanding of the gene's biology. Second, since three different transgenic phenotypes (sex ratio bias, all killing, no killing) have been found so far with only a few sequences analyzed, it is possible that further testing of new codons may increase the partial male killing to a full male-killing phenotype. Therefore, it would be fruitful and may be possible to continue to uncover a transgenic sequence that fully recapitulates the phenotype to refine this system as a study tool for *Wolbachia* male killing. Third and critically, although the results show that there is some relationship between synonymous codons and phenotype, several points remain for further testing. For example, we cannot conclude that the particular codon tested here is responsible for phenotype alterations in other host genetic backgrounds or species. It is possible that this codon plays a functional role only in a singular host genetic context. Here, we changed *wmk* sequences while holding the host genetic background fixed, but the reverse is required to conclude whether or not the particular codon plays a general role in other genotypes or natural contexts. Second, due to possible coevolution, various codons may or may not yield similar functional effects across different host backgrounds, and additional synonymous sites may contribute to the male-killing phenotype. Thus, the results here illuminate a previously unrecognized need for future research on the functional impacts of synonymous substitutions in endosymbionts. Future work may focus on determining if there is one specific synonymous codon that affects the male-killing function in all cases, if a more general feature exists where alteration of any or a subset of N-terminal or other *wmk* codons affects function, or if the effect of synonymous changes is specific to this background.

In addition, these findings are informative with regards to the more general study of phenotypes induced by endosymbiont transgenes. It is standard practice to codon optimize genes for maximizing host expression when testing endosymbiont gene function (*Perlmutter et al., 2019*; *LePage et al., 2017*; *Harumoto and Lemaitre, 2018*), with the assumption that synonymous codons will not alter function. However, if codon optimization potentially changes the interpretation of transgenic findings, then phenotypes should be corroborated in natural contexts once tools such as genetic editing are available in the relevant organism. Specifically, *wmk* should be knocked out in native contexts once it is more technically feasible. In addition, codon optimization algorithms are updated with new information periodically with the assumption that they yield improved results, although it is unclear in practice whether an algorithm is better optimized to produce results that reflect the true biology of a transgene. Future work is necessary to explore these concepts further. For example, comparisons of alleles may need to be performed with alleles identical in sequence except for any engineered differences, and the algorithm should remain constant across all transgenes that are compared to each other. Further, careful analysis and comparison of transgenic phenotypes produced by different algorithms may be necessary in some cases where the phenotype is known or expected. This approach could ensure the algorithm produces a transgenic phenotype that most closely resembles the natural phenotype. In addition, the reliability of certain molecular evolutionary signatures, such as dN/dS, may be compromised since synonymous mutations are assumed to be neutral in these analyses. These principles are particularly important for research on endosymbionts that increasingly relies on heterologous gene expression for functional studies. The smaller genomes of endosymbionts tend to have low GC content overall (*McCutcheon et al., 2009*), and GC content is the strongest driver of codon usage bias (*Knight et al., 2001*; *Chen et al., 2004*) and contributes to the strength of mRNA secondary structure (*Plotkin and Kudla, 2011*), so careful attention to the effects of synonymous changes may be of acute interest to the endosymbiont research community.

We also find interesting support for the hypothesis that there is co-adaptation between *wmk* homologs and their hosts since male-killing genes may be evolutionary matched for the host sex determination and molecular machinery that they manipulate. We first analyzed *wmk* homologs that are more distantly related to *w*Mel *wmk*, including from male-killing strains *w*Bol1b, *w*Bif, *w*Caub, and *w*Inn/*w*Bor (*Sasaki et al., 2002*; *Dyson et al., 2002*; *Dyer and Jaenike, 2004*; *Hurst et al., 2000*; *Sheeley and McAllister, 2009*). They did not recapitulate male killing when transgenically expressed in *D. melanogaster*. The lack of a sex ratio bias is expected if either *wmk* does not underpin male killing in these systems or divergent homologs are required to be closely co-adapted to their hosts. This latter hypothesis is based on observations that resistance to male killing is common (evidence of a potential host-microbe arms race) (*Charlat et al., 2005*; *Majerus and Majerus, 2010*; *Jaenike, 2007*; *Mitsuhashi et al., 2011*; *Hornett et al., 2008*), and some strains can cause male killing anew when transferred to another, usually closely-related host (*Jaenike, 2007*; *Fujii et al., 2001*; *Sasaki et al., 2002*) while not causing male killing in all recipient host species (*Veneti et al., 2012*; *Hughes and Rasgon, 2014*). It is possible, then, that the *wmk* homologs tested induce a sex ratio bias in their natural hosts, but not in *D. melanogaster*.

If we make the assumption that *wmk* is involved in male killing in nature, which requires confirmation beyond transgenic recapitulation of the phenotype, then the results here give the basis for additional hypotheses that require further testing. First, the killing of both males and females by the closely-related *w*Rec and *w*Suzi *wmk* homologs could indicate that the target is something shared in both males and females, but functions differently within each sex (assuming no off-target effects). Indeed, previous studies on *wmk* and *Wolbachia* male killing demonstrate a positive correlation with between dosage compensation complex (DCC) activity and DNA defects in male embryos (*Riparbelli et al., 2012*; *Perlmutter et al., 2019*; *Harumoto et al., 2018*). Notably, four of the five protein components of the DCC are expressed in both males and females, and it is only *msl-2* that is male-specific and catalyzes formation of the complex (*Lucchesi and Kuroda, 2015*). Thus, there are several non-sex-specific *wmk* target candidates that the gene product may interact with to cause lethality in males and females. Second, protein divergence and resultant conformational changes may impact the specificity between host target and the Wmk toxin, and could underlie development of host resistance. Male-killing genes are expected to evolve rapidly within hosts in order to counteradapt host resistance mechanisms and keep evolutionary pace with the rapid evolution of sex-related genes that may be the target of a male-killing toxin (*Marín and Baker, 1998*; *Rodriguez et al., 2007*). As such, major or minor divergence in protein or transcript sequence of either the host target or the microbial toxin may underpin changes that lead to common host resistance, *wmk*-host coadaptation, and functional transgene differences in the foreign *D. melanogaster* host. Third, these findings leave open the possibility of a variety of functionally relevant *wmk* protein or transcript conformations in nature, which could contribute to the marked diversity of *Wolbachia* male killing in terms of host species and sex determination systems (*Hurst et al., 1999*; *Charlat et al., 2005*; *Sasaki et al., 2002*; *Zeh et al., 2005*; *Dyer and Jaenike, 2004*; *Hurst et al., 2000*; *Fialho and Stevens, 2000*; *Van Borm et al., 2008*).

Taken together, this work reports previously unrecognized relationships for *wmk*-induced killing and establishes new hypotheses for the impacts of RNA structure and post-transcriptional processes in *wmk*-induced male killing. It also highlights several critical features for the research community regarding assumptions about the broad use of transgenes and the role of synonymous mutations in gene function. If *wmk* is involved in natural male killing, then this work could indicate how silent sequence changes may relate to known male-killing phenomena such as frequent host resistance or male-killing function in diverse hosts.

## Materials and methods

**Key resources table**

| Reagent type (species) or resource | Designation | Source or reference | Identifiers | Additional information |
|---|---|---|---|---|
| Gene (*Wolbachia pipientis*) | WD0626 | NCBI | NCBI:WD_RS02815 | Also known as *wmk* (*WO-mediated killing*) |

*Continued on next page*

*Continued*

| Reagent type (species) or resource | Designation | Source or reference | Identifiers | Additional information |
|---|---|---|---|---|
| Genetic reagent (*D. melanogaster*) | Act5c-Gal4/CyO | Bloomington *Drosophila* Stock Center | BDSC:3953; FlyBase FBti0012290 | P{AyGAL4}25 |
| Genetic reagent (*D. melanogaster*) | WT (y¹w⁶⁷ᶜ²³; P[CaryP]P2) | Bloomington *Drosophila* Stock Center | BDSC:8622; FlyBase FBti0040535 | WT strain used in this study; P{CaryP}attP2 |
| Genetic reagent (*D. melanogaster*) | *w*Mel *wmk* | This paper; *Perlmutter et al., 2019*; PMID:31504075 | | Expresses codon-optimized transgene; UAS promoter |
| Genetic reagent (*D. melanogaster*) | *w*Bol1b *wmk* | This paper | | Expresses codon-optimized transgene; UAS promoter |
| Genetic reagent (*D. melanogaster*) | *w*Bif *wmk* | This paper | | Expresses codon-optimized transgene; UAS promoter |
| Genetic reagent (*D. melanogaster*) | *w*Caub *wmk* | This paper | | Expresses codon-optimized transgene; UAS promoter |
| Genetic reagent (*D. melanogaster*) | *w*Inn/*w*Bor *wmk* | This paper | | Expresses codon-optimized transgene; UAS promoter; *w*Inn and *w*Bor *wmk* have same exact sequence |
| Genetic reagent (*D. melanogaster*) | *w*Suzi *wmk* | This paper | | Expresses codon-optimized transgene; UAS promoter |
| Genetic reagent (*D. melanogaster*) | *w*Rec *wmk* | This paper | | Expresses codon-optimized transgene; UAS promoter |
| Genetic reagent (*D. melanogaster*) | HA-*wmk* | This paper | | Expresses codon-optimized transgene; UAS promoter; 3 X HA tag epitope in linker between HTH domains of *w*Mel *wmk* |
| Genetic reagent (*D. melanogaster*) | 5′ *w*Rec *wmk* | This paper | | Expresses codon-optimized transgene; UAS promoter; Sequence has additional nine amino acids starting at upstream alternative start codon |
| Genetic reagent (*D. melanogaster*) | 5′ *w*Suzi *wmk* | This paper | | Expresses codon-optimized transgene; UAS promoter; Sequence has additional nine amino acids starting at upstream alternative start codon |
| Genetic reagent (*D. melanogaster*) | *w*Mel *wmk* (new) | This paper | | Expresses codon-optimized transgene; UAS promoter; Same exact sequence as *w*Mel *wmk*, newly transformed strain |
| Genetic reagent (*D. melanogaster*) | *w*Suzi codon | This paper | | Expresses codon-optimized transgene; UAS promoter; Same as *w*Mel *wmk*, but with 16ᵗʰ amino acid position using TCC Serine codon from *w*Suzi *wmk* strain |

*Continued on next page*

*Continued*

| Reagent type (species) or resource | Designation | Source or reference | Identifiers | Additional information |
|---|---|---|---|---|
| Genetic reagent (*D. melanogaster*) | *w*Rec codon | This paper | | Expresses codon-optimized transgene; UAS promoter; Same as *w*Mel *wmk*, but with 16th amino acid position using AGC Serine codon from *w*Rec *wmk* strain |
| Recombinant DNA reagent | pTIGER (plasmid) | *Ferguson et al., 2012*; PMID:22328499 | | Modified pUASp plasmid for enhanced germline expression under Gal4/UAS control |
| Sequence-based reagent | Rp49_F | This paper | PCR primers | CGGTTACGGAT CGAACAAGC |
| Sequence-based reagent | Rp49_R | This paper | PCR primers | CTTGCGCTTCT TGGAGGAGA |
| Sequence-based reagent | wmk_homologs_opt_F | This paper | PCR primers | CTGTATGCCATTG CCGAGACCCT |
| Sequence-based reagent | wmk_homologs_opt_R | This paper | PCR primers | TCACCAGATCCTTG GCGATCTTCATC |
| Sequence-based reagent | Msl-2_F | This paper | PCR primers | GGATTAACGCGGT CTAAGCATGTGTAACTG |
| Sequence-based reagent | Msl-2_R | This paper | PCR primers | GTATGCCGTCTG GGCCATGATG |
| Commercial assay or kit | Direct-zol RNA MiniPrep Kit | Zymo | R2051 | |
| Commercial assay or kit | Superscript VILO cDNA Synthesis Kit | ThermoFisher | 11754050 | |
| Chemical compound, drug | DNase, RNase-free | Ambion, Life Technologies | AM2222 | |
| Chemical compound, drug | iTaq Universal SYBR Green Mix | Bio-Rad | 1725120 | |
| Software, algorithm | GraphPad Prism 8 | GraphPad Prism 8 | RRID:SCR_002798 | |
| Software, algorithm | Geneious Pro v.2019.2; Geneious Pro v.2020.2.4 | Geneious | RRID:SCR_010519 | |
| Software, algorithm | jModelTest | jModelTest | RRID:SCR_015244 | |
| Software, algorithm | RNAfold WebServer | University of Vienna, *Gruber et al., 2008*, *Lorenz et al., 2011* | PMID:18424795; PMID:22115189 | http://rna.tbi.univie. ac.at/cgi-bin/RNAWeb Suite/RNAfold.cgi |

**Reagents source from this paper may be obtained from Bordenstein lab.

### *Drosophila* strains and maintenance

*D. melanogaster* strains used in this study include *Act5c*-Gal4/CyO (BDSC 3953, ubiquitously-expressing zygotic driver), the WT background line of genotype $y^1 w^{67c23}$; P[CaryP]P2 (BDSC 8622), the WD0626 (*wmk*) and WD0034 (control gene) transgene constructs previously described (*Perlmutter et al., 2019*), and several new transgene constructs. Briefly, each *Wolbachia* gene of interest was codon-optimized for optimal *D. melanogaster* expression using algorithms developed by GenScript Biotech (Piscataway, NJ). These sequences were then synthesized as DNA nucleotides and cloned by GenScript into the pTIGER plasmid (*Ferguson et al., 2012*). The pTIGER vector is a UASp-based plasmid optimized for germline expression and uses PhiC31 integrase (*Groth et al., 2004*) for targeted integration into the *D. melanogaster* genome. It also includes UAS promoters for inducible expression with a Gal4 driver (*Southall et al., 2008*) as well as a $w^+$ red eye marker for transformant screening. The plasmids were then sent to BestGene (Chino Hills, CA), which performed injections of the vectors into embryos of the BDSC 8622 background line with PhiC31 integrase to integrate the vector into the attP2 insertion site in the genome. Successful transformants were selected based on

the red eye marker, and each transgene line is descended from the offspring of a single transformant (isofemale).

*D. melanogaster* were reared on standard cornmeal, molasses, and yeast (CMY) media (5% w/v cornmeal (Quaker, Chicago IL), 1.875% w/v yeast (Red Star Yeast, Milwaulkee WI), 7.8% v/v molasses (Sweet Harvest Foods, Rosemount MN), 0.5% w/v type II *Drosophila* agar (Genesee Scientific, San Diego CA), 0.056% w/v tegosept (Genesee Scientific, San Diego CA), and 0.39% v/v propionic acid (Sigma Aldrich)). Stocks were maintained at 25 °C with virgin flies stored at room temperature. During virgin collections, stocks were kept at 18 °C overnight and 25 °C during the day. All flies were kept on a 12 hr light/dark cycle.

## Sex ratio assays

To assess the effect of transgene expression on adult sex ratios (measurement of male killing), sex ratio assays were performed as previously described (*Perlmutter et al., 2019*). Briefly, twenty biological replicates of 10 uninfected, 4- to 7-day-old virgin, female *Act5c*-Gal4/CyO driver flies and two uninfected, 1- to 2-day-old virgin, male transgene flies were each set up in vials with CMY media. Individuals were randomly allocated to each vial after all females or males of a given genotype were mixed together. They were left on the media to lay eggs for 4 days at 25 °C with a 12 hr light/dark cycle, at which point adults were discarded. The vials are then left at 25 °C until the offspring are counted. After 9 days of adult offspring emergence, they were scored for both sex and expression (red eye color from *Act5c*-Gal4 chromosome) or non-expression (curly wings from CyO balancer chromosome). The number of adult offspring per vial across all experiments ranges from 50 to 170, with a mean of 120 (ranges and means per experiment are included in figure captions). Any vials with fewer than 50 adult offspring were removed from the analysis, as this indicates either abnormally poor egg laying or hatching (typically 0–2 vials per group). In addition, vials with no adult emergence, while others of the same genotype had typical levels of offspring, were also excluded (typically 2–3 vials per group). Results were graphed in GraphPad Prism 8.4.0, which applies a 'Standard Plot Appearance' correction for visibility of data distribution where the width of distribution of points is proportional to the number of points at that y-value.

## RNA secondary structures

RNA secondary structures were generated by uploading the nucleotide sequences of the indicated gene to the RNA fold web server (*Gruber et al., 2008*; *Lorenz et al., 2011*). The structures shown are the graphical outputs of the MFE (minimum free energy) secondary structures. Colors indicate base pair probabilities, from blue to red, with blue indicating a probability of 0 and red indicating a probability of 1.

## Gene expression

Gene expression was measured in *Drosophila* embryos aged 4–5 hr AED. Each point represents a biological replicate with the RNA of 30 pooled embryos from crosses between a unique set of 60 uninfected, 4- to 7-day-old virgin, female *Act5c*-Gal4/CyO driver flies and 12 uninfected, 1- to 2-day-old virgin, male transgene flies of the indicated genotype. Each point represents a biological replicate from different bottles. Individuals were randomly allocated to each bottle after all females or males of a given genotype were mixed together. Each collection chamber consisted of a grape juice agar plate with yeast in an eight oz round bottom bottle, and flies. These were placed in a 25 °C incubator overnight (16 hr). Then, the plates were swapped with fresh ones. The flies were allowed to lay eggs for 1 hr. The plates were then left at 25 °C for an additional 4 hr to age them to be 4–5 hr old (the estimated time of male death in *wmk* crosses). Embryos were then gathered in groups of 30 (each group from a unique bottle/biological replicate) and flash frozen in liquid nitrogen. RNA was extracted using the Direct-zol RNA MiniPrep Kit (Zymo), RNase-free DNase (Ambion, Life Technologies), cDNA was generated with SuperScript VILO (Invitrogen), and RT-qPCR was run using iTaq Universal SYBR Green Mix (Bio-Rad). qPCR was performed on a Bio-Rad CFX-96 Real-Time System. Primers are listed in the Key Resources Table. Conditions were as follows: 50 °C 10 min, 95 °C 5 min, 40 x (95 °C 10 s, 55 °C 30 s), 95 °C 30 s. Differences in gene expression were done by calculating $2^{-\Delta ct}$ (difference in ct values of two genes of interest). Data points were excluded if a sample had low-quality cDNA that did not

amplify in qPCR. Data points of each biological replicate are measured as the mean of two technical replicates from each sample.

## Phylogenetic trees

The nucleotide phylogenetic trees of host COI genes and *wmk* native gene or transgene sequences were inferred based on a MUSCLE alignment in Geneious Prime 2020.2.4 followed by stripping all sites with gaps. The resulting 652, 690, 686 nucleotide base pair alignments (respectively) were then analyzed via jModelTest 2.1.10 v20160303 (*Darriba et al., 2012Guindon and Gascuel, 2003*). The AICc-corrected best model, JC, was predicted for all three alignments and was used to build the trees using the MrBayes (*Huelsenbeck and Ronquist, 2001*; *Ronquist and Huelsenbeck, 2003*) Geneious plugin with the JC69 model (*Jukes and Cantor, 1969*) and equal rate variation.

## Transgene sequence alignments

The sequence alignments of different *wmk* transgenes in *Figure 2* and *Figure 4* were conducted in Geneious Pro v.2019.2 using a MUSCLE alignment. Black bars in *Figure 2* indicate sequence mismatches compared to the *w*Mel *wmk* transgene reference sequence with any gaps stripped. Codons in *Figure 4* are colored by amino acid.

## Statistical analyses

Sample sizes for experiments were based on previous publications, which demonstrated repeatability in relative differences between treatment groups. Each experiment was completed twice, and statistical tests were applied to both to confirm repetition in differences between treatment groups. In all plots, the first experiment is the representative one shown. For sex ratios, we tested different sample sizes for reliability in sex ratio measurements, and found that 20 biological samples per group, with 10 females and 2 males per sample, resulted in consistently replicable data, which is the standard we apply here. qPCR data was approached similarly, by previous work in the lab demonstrating that with embryos 4–5 h AED, we are able to get consistent, high-quality, replicable data with 30 embryos per sample, and at least eight samples per group.

All statistical analyses for sex ratios and were performed using GraphPad Prism eight software. For sex ratios, a non-parametric Kruskal-Wallis one-way ANOVA followed by Dunn's multiple comparisons test was applied to all gene-expressing categories, followed by the same test but on all non-expressing categories. For gene expression, groups were compared using the same Kruskal-Wallis one-way ANOVA with Dunn's multiple comparisons test. Significant results are indicated with * symbols in figures, with accompanying p values in captions. Any comparisons with no symbol are nonsignificant. Full statistical information and outputs for all sex ratio and qPCR data is available in the Source Data file.

## Acknowledgements

This work was supported by National Institutes of Health (NIH) grant R21 AI133522 to SRB, the Vanderbilt Microbiome Innovation Center, and NIH grants F31 AI143152, K-INBRE P20 GM103418, and National Science Foundation (NSF) Postdoctoral Research Fellowship (PRFB) DBI 2109772 to JIP. We would also like to thank RL Unckless for helpful comments on phylogenies.

## Additional information

### Competing interests

Jessamyn I Perlmutter, Seth R Bordenstein: Is listed as an author on a patent related to the use of wmk in vector control. US Patent 20210000092 16/982708. The other author declares that no competing interests exist.

## Funding

| Funder | Grant reference number | Author |
|---|---|---|
| National Institutes of Health | R21 AI133522 | Seth R Bordenstein |
| National Institutes of Health | F31 AI143152 | Jessamyn I Perlmutter |
| Vanderbilt Microbiome Innovation Center | General Funds | Seth R Bordenstein |
| National Institutes of Health | P20 GM103418 | Jessamyn I Perlmutter |
| National Science Foundation | DBI 2109772 | Jessamyn I Perlmutter |

The funders had no role in study design, data collection and interpretation, or the decision to submit the work for publication.

### Author contributions

Jessamyn I Perlmutter, Conceptualization, Data curation, Formal analysis, Funding acquisition, Investigation, Methodology, Supervision, Visualization, Writing – original draft, Writing – review and editing; Jane E Meyers, Investigation, Writing – original draft; Seth R Bordenstein, Conceptualization, Funding acquisition, Methodology, Project administration, Resources, Supervision, Writing – original draft, Writing – review and editing

### Author ORCIDs

Jessamyn I Perlmutter ⓘ http://orcid.org/0000-0002-9789-4674
Seth R Bordenstein ⓘ http://orcid.org/0000-0001-7346-0954

### Decision letter and Author response

Decision letter https://doi.org/10.7554/eLife.67686.sa1
Author response https://doi.org/10.7554/eLife.67686.sa2

## Additional files

### Supplementary files

- Transparent reporting form
- Source data 1. Source data for all graphical data sets and statistical tests performed for this study.

### Data availability

All data generated or analyzed during this study are included in the manuscript and supporting files. Source data files have been provided for Figures 3-6.

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
