## [Editor Report]

This study identified the genetic mechanisms underlying sex-ratio distortion through male-killing in *Drosophila melanogaster* flies infected with the endosymbiont Wolbachia. The endosymbiont carries the prophage WO, which is the center of interest in this study. The key result of this study is that a synonymous mutation in a prophage gene can explain the differences between sex-ratio distorting and not distorting symbionts. The finding, that a synonymous SNP plays a key role is not entirely novel in biology, but there are only few examples known of this type of genotype–phenotype association.

---

## [Decision Letter]

**Decision letter after peer review:**

Thank you for submitting your article "A single synonymous nucleotide change impacts the male-killing phenotype of prophage WO gene wmk" for consideration by *eLife*. Your article has been reviewed by 2 peer reviewers, one of whom is member of our Board of Reviewing Editors, and the evaluation has been overseen by Patricia Wittkopp as the Senior Editor. The reviewers have opted to remain anonymous.

Essential revisions:

1) The most important comment from the review is that it is plausible that the observed changes in the effect and strength of killing are due to an interaction between host and wmk genotype. This has implications for unravelling the underlying genetic basis to the male-killing phenotype more widely. Therefore, it will be critical to repeat some of the main findings in other *D. melanogaster* genotypes to determine the importance of the variation in the wmk homologs more generally.

2) The manuscript requires extensive streamlining. The text is hard to follow and has too many details. The main points are often hidden among details. A more focussed and shortened manuscript would be highly welcome.

*Reviewer #1:*

This study aims to find the genetic mechanisms underlying sex-ratio distortion through male-killing in *Drosophila melanogaster* flies infected with the endosymbiont Wolbachia. The endosymbiont carries the prophage WO, which is in the center of interested in this study. The key result of this study is that a synonymous mutation in a prophage gene can explain the differences between sex-ratio distorting and not distorting symbionts. The study uses transgene technology to modify phage genes and to investigate which changes in the gene is involved in the phenotype. The finding, that a synonymous SNP plays a key role is not entirely novel in biology, but there are only few examples known of this type of genotype–phenotype association. The study does not include experiments to show that the main finding is not limited to one particular background of the fly line used. An experiment including multiple genotypes would be needed to show this.

The study is mostly clear and easy to follow, but requires a lot of attention. The authors choose to build up the story as I guess it was carried out in the lab. Thus, the reader is guided through every step of the process. While I see that this is appealing from the way the study was carried out, it results in a very long manuscript with a lot of material that would be much better placed in a supplement.

The introduction seems unfocused. It meanders around, jumping from topic to topic and does not give the reader a sense of where things will go. Figure 1 gives an overview about the different aspects addressed here, but it is not used to guide the reader through the different lines of thought addressed in the introduction. If Figure 1 will stay (I actually think it is not needed) it should be introduced earlier and used as a road map for the paper. Alternatively, the introduction could stay more general and only in the last paragraph the different ways the system is studied will be summarized. Along these lines, it would be good to have a better reasoning for the combination of experiments conducted. It is left to the reader to understand why certain types of experiments have been done. On the other hand, the introduction misses a section on the biology of the phage and its interaction with the host(s). It is hard to understand the biology of the system without getting an understanding of the insect – Wolbachia – phage interactions. For non-specialist, understanding the role of the three players is essential for the system.

The result section could be easily shortened by focusing on the essential experiments. Experiments that do not contribute to the final result can go into the supplement.

Also the discussion is much too long. I suggest to reduce it to half and focus on the important points and the take-home messages. Currently the discussion follows the way the results are presented in the result section. However, this is not needed. The important finding should be discussed first. Findings that are important in the development of the project, may not be important for the biology of the system overall. And they may not be important for the reader.

The text contains many abbreviations. For less specialist readers this become rather difficult. I suggest to reduce abbreviations and to add a list with them, so one can find them fast. All abbreviations have to be explained when they are used for the first time. Furthermore, abbreviations are used in different forms making part of the text or figures messy. As an example, look at Figure 4: going plot by plot from A to E the names of the treatments changes from plot to plot. This is issues persists throughout text and figures of the entire manuscript.

On line 158-160 it is said that wSuzi differs only by one SNP from wMel. On lines 162-164 it is then said that the transgenes were codon optimized. Why is this necessary, if only 1 position differs? More generally: " codon-optimization " plays an important role in this study. Please explain why codon-optimization was done and what was exactly done.

For several experiment it is said "The experiment has been performed twice". How where the data treated? The stats described do not allow a complex design with repeated experiment. Did you pool the data of the two experiments? Did you test if the outcome of the experiments differed?

Line 241: "Thus, we conclude that small or large wmk transcript sequence changes can lead to alterations in predicted RNA structure or possibly protein structure that may relate to altered phenotypes in wmk homologs." This is a very vague conclusion. It basically says nothing.

For the plots of the individual data points a jitter function was used (please say so!). Was the function used to jitter in both dimensions or only in the X-axis dimension?

Animals were kept on a standard media. Please give references when you make such statements.

The methods section regarding the transgene constructs is rather short. I would like to see more information to understand what was done, why it was done and how it was done.

*Reviewer #2:*

This study aims to unravel the genomic basis to wmk-induced male killing by transgenically expressing homologs of varying relatedness, with synonymous nucleotide changes, and predicted alternative start codons in *D. melanogaster* flies. The study builds on previous work showing that expression of wmk in fly embryos recapitulates several aspects of male killing. While more distantly-related homologs did not induce male killing when expressed in *D. melanogaster*, more closely-related wmk homologs induce either killing of both sexes or male killing only. However, the male-killing phenotype was not due to amino acid differences, but associated with RNA structural differences of the different wmk homologs. In addition, only one synonymous nucleotide change was sufficient to ablate the killing phenotype. These findings suggests that minor and even silent nucleotide differences impact on the expression of male killing in *D. melanogaster*. It is concluded that a new model incorporating the impacts of RNA structure and post-transcriptional processes in wmk-induced male killing needs to be developed.

The strength of the study lies in the systematic and carefully controlled approach to quantify the phenotypic effects of both sequence and structural changes to various wmk homologs for inducing the male-killing phenotype. Detailed dissection of the phenotypic impact of minor changes to the wmk homologs including sequence variation, silent nucleotide changes, and RNA structural differences was quantified. This approach reveals a complex genotype-phenotype relationship, but highlights the importance of including post-translational processes. The data is novel in that previous work have largely ignored structural changes and assumed that synonymous differences in codons has no effect on protein function, whereas the current study based on updated codon optimization algorithms reveal that this assumption is incorrect. The finding highlights the importance of considering also structural genetic variation for phenotypic expression differences. This suggestion is further corroborated by the lack of difference in wmk homologue expression levels, indicating that the functional differences are due to post-translational effects.

There are limitations to the findings of this complex genotype-phenotype relationship. The current study only examined the phenotypic impact by expressing the different homologs in one *D. melanogaster* genetic background. Given the variability of the phenotypic pattern revealed based on minor changes to the wmk homologs, it will be critical to repeat some of the main findings in other *D. melanogaster* genotypes to determine the importance of the variation in the wmk homologs more generally. It is entirely plausible that the observed changes in the effect and strength of killing is due to an interaction between host and wmk genotype. This has implications for unravelling the underlying genetic basis to the male-killing phenotype more widely. It is as yet to be demonstrated whether wmk is involved in male killing in natural population, and to what extent there are shared patterns and mechanisms of male killing induced by other bacterial endosymbionts such as Spiroplasma.

I found the study of interest and enjoyed reading this manuscript. Overall, it represents a careful dissection of the functional genomic basis of the wmk candidate gene for the male-killing phenotype in *D. melanogaster*. The experimental design and choice of the putative male-killing wmk homologs of varying relatedness provides an opportunity to examine the importance of genetic similarity, in conjunction with wmk-variants with synonymous nucleotide changes, and predicted alternative start codons for the male killing expression. The complex pattern of genotype-phenotype relationship that is unravelled is perhaps unexpected, but possibly more important is the finding that RNA structural differences are associated with male killing. This finding has several implications such as the need to consider the importance of post-translational processes, but also that the assumption that synonymous differences in codons has no effect on protein function is erroneous. On the whole these complex patterns suggest that we need to re-visit the model for the genotype-phenotype relationship of male killing. As such, the study makes a valuable contribution to the field.

However, the study also has some shortcomings and limitations. Importantly, the impact of the various wmk homologs were only examined in one *D. melanogaster* genotype. Given the variability of the phenotypic pattern based on minor changes to the wmk homologs that was unravelled, it will be critical to repeat the main findings in other *D. melanogaster* genotypes to determine the importance of the variation in the wmk homologs for male killing irrespective of host genetic background. The suggestion that caution need to be taken when examining sequence variation and in particular synonymous nucleotide changes as their impact may be dependent on RNA structure is also based on few observations.

In general, little consideration is made to the evolutionary implications of the finding. It is currently unclear what patterns are to be expected in other host-gene combinations and as such the findings have perhaps low predictive power. It is entirely plausible that the observed changes in the effect and strength of killing is due to the interaction between host and wmk genotype shaped by co-evolution. This caveat has implications for understanding the underlying genetic basis to male-killing more widely, especially since one of the aims of the study was to address the genotype-phenotype relationship that underpins wmk function. Here the wmk genotype is varied while keeping the host genotype constant. This inevitably limit the generality of the findings, especially as small wmk differences were associated with big phenotypic differences. More work is therefore required to address the genotype-phenotype relationship by expressing some of the wmk homologs across different *D. melanogaster* genotypes.

Overall, while I found the discussion interesting, it is currently highly speculative and perhaps some of the suggestions that are based on limited amount of supporting data could be toned down. For example, the finding that only one synonymous nucleotide change was sufficient to remove the killing phenotype is intriguing. However, it is not clear if the suggestion made is that resistance to male killing is easily achieved by minor changes to wmk that are functionally dependent on codon biases or if minor changes in general can have large phenotypic effects. If correctly interpreted, then more *D. melanogaster* genotypes need to be assessed to verify this suggestion, and if resistance is achieved this way, then outlining some specific predictions about the expected variation in resistance strains (i.e. non-synonymous or structural) would be valuable.

It is unclear how robust the updated codon optimization algorithms are regarding how accurately they predict protein structure and function. It is noticeable that significant differences were found using the updated codon optimization algorithm compared to the older one from four years ago. How certain can we be that the new algorithm better reflect protein structure and function as it is rightly stated that the codon optimization may change the interpretation of results from transgenic experiments. The caution raised is valid, but perhaps it would be valuable to provide some suggestion regarding how to best mitigate this potential issue.

Finally, it is yet to be demonstrated whether wmk is involved in male killing also in natural population, and to what extent there are shared patterns and mechanisms of male killing induced by other bacterial endosymbionts such as Spiroplasma.

---

## [Author Response]

Essential revisions:1) The most important comment from the review is that it is plausible that the observed changes in the effect and strength of killing are due to an interaction between host and wmk genotype. This has implications for unravelling the underlying genetic basis to the male-killing phenotype more widely. Therefore, it will be critical to repeat some of the main findings in other *D. melanogaster* genotypes to determine the importance of the variation in the wmk homologs more generally.

We understand the importance of this point and address it in more detail below in the response to the first comment by Reviewer 1. We would also like to thank the Editor and Reviewers for considering and acknowledging the request to not require this experiment for the reasons outlined below. Per the follow-up recommendation, we now acknowledge the limitations of examining one genetic background in the discussion and altered language to avoid perceptions of generalizability. In individual responses to reviewer comments below, we included some more detailed explanation and example text.

2) The manuscript requires extensive streamlining. The text is hard to follow and has too many details. The main points are often hidden among details. A more focussed and shortened manuscript would be highly welcome.

Thank you for this feedback. We agree and removed unnecessary elements in various sections, moved some parts of the results to the supplement, removed any part of the results that were redundant with other parts of the paper, and reordered and streamlined the discussion as well. The discussion is now a few pages shorter, as are the results. Specific changes are outlined where reviewers suggested them below.

Reviewer #1:This study aims to find the genetic mechanisms underlying sex-ratio distortion through male-killing in *Drosophila melanogaster* flies infected with the endosymbiont Wolbachia. The endosymbiont carries the prophage WO, which is in the center of interested in this study. The key result of this study is that a synonymous mutation in a prophage gene can explain the differences between sex-ratio distorting and not distorting symbionts. The study uses transgene technology to modify phage genes and to investigate which changes in the gene is involved in the phenotype. The finding, that a synonymous SNP plays a key role is not entirely novel in biology, but there are only few examples known of this type of genotype –phenotype association. The study does not include experiments to show that the main finding is not limited to one particular background of the fly line used. An experiment including multiple genotypes would be needed to show this.

We agree that recapitulating the results in other backgrounds is intriguing and important for establishing a broader role of these findings. We thank the Reviewers and Editor for allowing us to pursue this line of investigation separately from this work, and we now discuss what experiments can be completed to answer these and other questions. We also edited the manuscript to tone down any conclusions that would imply generalizability of the findings at this point. For example:

“For example, we cannot conclude that the particular codon tested here is responsible for phenotype alterations in other host genetic backgrounds or species. It is possible that this codon plays a functional role only in a singular host genetic context. Here, we changed wmk sequences while holding the host genetic background fixed, but the reverse is required to conclude whether or not the particular codon plays a general role in other genotypes or natural contexts. Second, due to possible coevolution, various codons may or may not yield similar functional effects across different host backgrounds, and additional synonymous sites may contribute to the male-killing phenotype. Thus, the results here illuminate a previously unrecognized need for future research on the functional impacts of synonymous substitutions in endosymbionts. Future work may focus on determining if there is one specific synonymous codon that affects the male-killing function in all cases, if a more general feature exists where alteration of any or a subset of N-terminal or other wmk codons affects function, or if the effect of synonymous changes is specific to this background.”

Text summarizing the 06/21/2021 query to the Editor and Reviewers for further clarification: We believe there are several reasons why the results can stand on their own, while appropriately acknowledging caveats. First, we note the lack of genetic background testing on previous transgene experiments driving the major discoveries of *Wolbachia* genes involved in reproductive parasitism. This requirement would therefore hold the current work to a novel bar not previously applied by the field. In addition, the genetic background here is the same as used in previous work on these phenotypes, making it the most pertinent to test and inform previous and ongoing studies by many research groups. Second, the results shown here would still stand no matter the results of genetic background testing and would demonstrate that it is possible for synonymous changes to have functional relevance in the transgenic *wmk* phenotype. The major findings are still novel in the field, relevant to ongoing studies of reproductive parasitism, and informative regarding one of the most common genetic backgrounds. Finally, we note that two different lines with unique synonymous codon changes (the final experiment) independently created the same result that a synonymous codon change ablates phenotype, providing additional robustness to our findings. Doing additional experiments would be logistically difficult. Barriers include the relocation of the first author of the work to another lab for a postdoctoral position, completion of the funding for the project, remaining institutional COVID-19 restrictions, and lack of replacement personnel in the lab to continue the work. Notably, there is also the non-trivial requirement to create and test new transgene lines that would be costly and take nearly a year to complete (the experiments in the manuscript already took several years and the new fly lines would cost thousands to make).

The study is mostly clear and easy to follow, but requires a lot of attention. The authors choose to build up the story as I guess it was carried out in the lab. Thus, the reader is guided through every step of the process. While I see that this is appealing from the way the study was carried out, it results in a very long manuscript with a lot of material that would be much better placed in a supplement.

We thank the reviewer for pointing this out. We shortened the manuscript by removing redundant information and transferring some parts of the results to the supplement. We also removed about three pages of text from the discussion (before adding in new sections as requested by reviewers).

The introduction seems unfocused. It meanders around, jumping from topic to topic and does not give the reader a sense of where things will go.

We added a few topics into the Introduction as recommended in other comments, and we edited various portions of the Introduction to connect the ideas together more clearly. We hope the changes are now satisfactory, and we are of course happy to consider further feedback.

Figure 1 gives an overview about the different aspects addressed here, but it is not used to guide the reader through the different lines of thought addressed in the introduction. If Figure 1 will stay (I actually think it is not needed) it should be introduced earlier and used as a road map for the paper. Alternatively, the introduction could stay more general and only in the last paragraph the different ways the system is studied will be summarized.

We edited the final paragraph of the Introduction to more comprehensively cover the content of the figure and full direction of the paper. For readers not familiar with the biological system or questions, we believe this figure will serve as a gateway to the genetic alterations conducted in the experiments.

Along these lines, it would be good to have a better reasoning for the combination of experiments conducted. It is left to the reader to understand why certain types of experiments have been done.

It was not clear to us at the outset of these experiments what results would ultimately emerge and what follow-up experiments would be necessary as our initial hypotheses were proven wrong with many of the surprises from the work. So, there was no *a priori* reasoning for why experiments were done until we had the results of the previous experiments. We agree that this makes the reading a bit confusing. As such, we clarified the logic flow in the Results section as the narrative progresses from experiment to experiment, and we reorganized some of the introduction to improve transition statements and offer a roadmap to readers earlier on.

On the other hand, the introduction misses a section on the biology of the phage and its interaction with the host(s). It is hard to understand the biology of the system without getting an understanding of the insect – Wolbachia – phage interactions. For non-specialist, understanding the role of the three players is essential for the system.

Thank you for the suggestion. We now add a section introducing phage WO and its relevance to the phenotypes tested here.

“The wmk gene and two cytoplasmic incompatibility factor (cif) genes that underlie cytoplasmic incompatibility (a parasitism phenotype whereby offspring die in crosses between infected males and uninfected females) occur in the eukaryotic association module (EAM) of prophage WO, which refers to the phage WO genome that is inserted into the bacterial chromosome. The EAM is common in WO phages across several Wolbachia strains and is rich in genes that are homologous to eukaryotic genes or annotated with eukaryotic functions. As such, the expression of reproductive parasitism genes from the EAM and tripartite interactions between phage WO, Wolbachia, and eukaryotic hosts are central to Wolbachia’s ability to interact with and modify host reproduction.”

The result section could be easily shortened by focusing on the essential experiments. Experiments that do not contribute to the final result can go into the supplement.

We removed redundant sentences and made some figures supplemental.

Also the discussion is much too long. I suggest to reduce it to half and focus on the important points and the take-home messages. Currently the discussion follows the way the results are presented in the result section. However, this is not needed. The important finding should be discussed first. Findings that are important in the development of the project, may not be important for the biology of the system overall. And they may not be important for the reader.

We reordered the discussion to cover the biggest findings first, and removed about a third of the original writing in the discussion.

The text contains many abbreviations. For less specialist readers this become rather difficult. I suggest to reduce abbreviations and to add a list with them, so one can find them fast. All abbreviations have to be explained when they are used for the first time. Furthermore, abbreviations are used in different forms making part of the text or figures messy. As an example, look at Figure 4: going plot by plot from A to E the names of the treatments changes from plot to plot. This is issues persists throughout text and figures of the entire manuscript.

Thank you for pointing this out. In the manuscript, we used labels such as “*w*Mel” or “*w*Mel *wmk*” synonymously. Since that resulted in too many labels and acronyms to follow and sift through, we unified the terminology throughout by ensuring that the same label for each genotype remains constant in the text, and we edited the figures to ensure that the terminology is kept constant. For example, we now refer to the original homolog exclusively as “*w*Mel *wmk*” as opposed to using this term in some cases and “*wmk*” in others.

On line 158-160 it is said that wSuzi differs only by one SNP from wMel. On lines 162-164 it is then said that the transgenes were codon optimized. Why is this necessary, if only 1 position differs? More generally: " codon-optimization " plays an important role in this study. Please explain why codon-optimization was done and what was exactly done.

We added a short explanation to the first figure to explain the codon optimization to readers, but here is an additional, expanded explanation for reference: Codon-optimization is standard for transgenic expression analyses since the DNA sequence comes from an organism (bacteria) that is divergent compared to its destination organism (flies). For synonymous codons, although all code for the same amino acid, each organism has a specific bias for which tRNAs are more or less abundant. Synonymous codon differences can result in variations in RNA folding (based on subtle molecular differences in the codons) as well as differences in translation such as translation rate (based on codon availability in terms of rarity). Expressing a bacterial gene in a fly host without optimizing the codons would mean that the gene would likely contain many codons that are not preferred/as common in the fly host, and this could result in altered translation rates or even early termination when the fly expresses the gene. For this reason, we and others utilizing transgenics take all the *Wolbachia* sequences and put them through an algorithm that gives the same protein sequence, but with the best codon combinations for optimal fly expression. This new “codon-optimized” sequence is then utilized to synthesize the gene anew for insertion into the fly host.

In the specific lines referenced here, we clarify that while the bacterial sequences differ by a single SNP, they must be codon optimized for fly expression for the reasons explained above, which then of course changes the DNA sequence. Since this is an important principle that we want readers to clearly understand before going into the results, we added a new section to the Introduction to introduce the concept of codon optimization and transgenes. We also expanded the Discussion of this topic as well. The Introduction change is noted below:

“These sequences are codon-optimized based on different codon biases due to different tRNA abundances in the divergent bacterial source and eukaryotic destination species.”

For several experiment it is said "The experiment has been performed twice". How where the data treated? The stats described do not allow a complex design with repeated experiment. Did you pool the data of the two experiments? Did you test if the outcome of the experiments differed?

In each case, experiments were done twice to confirm that the results were repeatable. We applied the same statistical tests to each individual experiment, and we determined that they both showed the same results. However, the data from each individual experiment were not combined. This is because exact y-values often differ from experiment to experiment due to small, uncontrollable environmental factors, even if relative differences between treatment groups are repeatable. This is typical of *Drosophila* egg-laying experiments, where the average number of eggs laid naturally varies within a wide range due to sensitivity to environmental factors and gives minor differences in results. The large sample size and sample size cutoff values were chosen because they largely correct for these small variations and give consistent results relative to groups within each experiment. However, there is still some variation in absolute values from experiment to experiment, so we did not combine data. For the data in the manuscript, all plots were from the first iteration of the experiment as a representative result of the two. Brief statements on this have been added to the methods section.

Line 241: "Thus, we conclude that small or large wmk transcript sequence changes can lead to alterations in predicted RNA structure or possibly protein structure that may relate to altered phenotypes in wmk homologs." This is a very vague conclusion. It basically says nothing.

The statement has been deleted.

For the plots of the individual data points a jitter function was used (please say so!). Was the function used to jitter in both dimensions or only in the X-axis dimension?

We graphed these plots using GraphPad Prism software, which does apply a function to horizontally distribute points at the same y-value evenly across for greater visibility of data distribution. We have now included a statement regarding this in the methods, including which particular function was used.

Animals were kept on a standard media. Please give references when you make such statements.

We used standard CMY media, but recipes may vary slightly lab to lab. For clarity, we added details of our specific recipe in the methods section.

The methods section regarding the transgene constructs is rather short. I would like to see more information to understand what was done, why it was done and how it was done.

We expanded the Methods section to include additional details on making *Drosophila* transformants so it can be repeated. The text is copied here below:

“*D. melanogaster* strains used in this study include Act5c-Gal4/CyO (BDSC 3953, ubiquitously-expressing zygotic driver), the WT background line of genotype y^1^w^67c23^; P[CaryP]P2 (BDSC 8622), the WD0626 (wmk) and WD0034 (control gene) transgene constructs previously described^25^, and several new transgene constructs. […] The plasmids were then sent to BestGene (Chino Hills, CA), which performed injections of the vectors into embryos of the BDSC 8622 background line with PhiC31 integrase to integrate the vector into the attP2 insertion site in the genome. Successful transformants were selected based on the red eye marker, and each transgene line is descended from the offspring of a single transformant (isofemale).”

Reviewer #2:[…] There are limitations to the findings of this complex genotype-phenotype relationship. The current study only examined the phenotypic impact by expressing the different homologs in one *D. melanogaster* genetic background. Given the variability of the phenotypic pattern revealed based on minor changes to the wmk homologs, it will be critical to repeat some of the main findings in other *D. melanogaster* genotypes to determine the importance of the variation in the wmk homologs more generally. It is entirely plausible that the observed changes in the effect and strength of killing is due to an interaction between host and wmk genotype. This has implications for unravelling the underlying genetic basis to the male-killing phenotype more widely. It is as yet to be demonstrated whether wmk is involved in male killing in natural population, and to what extent there are shared patterns and mechanisms of male killing induced by other bacterial endosymbionts such as Spiroplasma.

We addressed this point in more detail above in the first response to the comments from Reviewer 1.

I found the study of interest and enjoyed reading this manuscript. Overall, it represents a careful dissection of the functional genomic basis of the wmk candidate gene for the male-killing phenotype in *D. melanogaster*. The experimental design and choice of the putative male-killing wmk homologs of varying relatedness provides an opportunity to examine the importance of genetic similarity, in conjunction with wmk-variants with synonymous nucleotide changes, and predicted alternative start codons for the male killing expression. The complex pattern of genotype-phenotype relationship that is unravelled is perhaps unexpected, but possibly more important is the finding that RNA structural differences are associated with male killing. This finding has several implications such as the need to consider the importance of post-translational processes, but also that the assumption that synonymous differences in codons has no effect on protein function is erroneous. On the whole these complex patterns suggest that we need to re-visit the model for the genotype-phenotype relationship of male killing. As such, the study makes a valuable contribution to the field.However, the study also has some shortcomings and limitations. Importantly, the impact of the various wmk homologs were only examined in one *D. melanogaster* genotype. Given the variability of the phenotypic pattern based on minor changes to the wmk homologs that was unravelled, it will be critical to repeat the main findings in other *D. melanogaster* genotypes to determine the importance of the variation in the wmk homologs for male killing irrespective of host genetic background. The suggestion that caution need to be taken when examining sequence variation and in particular synonymous nucleotide changes as their impact may be dependent on RNA structure is also based on few observations.

We addressed this point in more detail above in the first response to the comments from Reviewer 1.

In general, little consideration is made to the evolutionary implications of the finding. It is currently unclear what patterns are to be expected in other host-gene combinations and as such the findings have perhaps low predictive power. It is entirely plausible that the observed changes in the effect and strength of killing is due to the interaction between host and wmk genotype shaped by co-evolution. This caveat has implications for understanding the underlying genetic basis to male-killing more widely, especially since one of the aims of the study was to address the genotype-phenotype relationship that underpins wmk function. Here the wmk genotype is varied while keeping the host genotype constant. This inevitably limit the generality of the findings, especially as small wmk differences were associated with big phenotypic differences. More work is therefore required to address the genotype-phenotype relationship by expressing some of the wmk homologs across different *D. melanogaster* genotypes.

We edited the Discussion to reflect the reviewer’s concern, such as the statement below:

“Third and critically, although the results show that there is some relationship between synonymous codons and phenotype, several points remain for further testing. […] Here, we changed wmk sequences while holding the host genetic background fixed, but the reverse is required to conclude whether or not the particular codon plays a general role in other genotypes or natural contexts."

Overall, while I found the discussion interesting, it is currently highly speculative and perhaps some of the suggestions that are based on limited amount of supporting data could be toned down. For example, the finding that only one synonymous nucleotide change was sufficient to remove the killing phenotype is intriguing. However, it is not clear if the suggestion made is that resistance to male killing is easily achieved by minor changes to wmk that are functionally dependent on codon biases or if minor changes in general can have large phenotypic effects. If correctly interpreted, then more *D. melanogaster* genotypes need to be assessed to verify this suggestion, and if resistance is achieved this way, then outlining some specific predictions about the expected variation in resistance strains (i.e. non-synonymous or structural) would be valuable.

We edited the Discussion to include clearer caveats and statements on the limitations of our study, as well as what remains to be determined in future work. An example is included below.

“Second, due to possible coevolution, various codons may or may not yield similar functional effects across different host backgrounds, and additional synonymous sites may contribute to the male-killing phenotype. […] Future work may focus on determining if there is one specific synonymous codon that affects the male-killing function in all cases, if a more general feature exists where alteration of any or a subset of N-terminal or other wmk codons affects function, or if the effect of synonymous changes is specific to this background.”

It is unclear how robust the updated codon optimization algorithms are regarding how accurately they predict protein structure and function. It is noticeable that significant differences were found using the updated codon optimization algorithm compared to the older one from four years ago. How certain can we be that the new algorithm better reflect protein structure and function as it is rightly stated that the codon optimization may change the interpretation of results from transgenic experiments. The caution raised is valid, but perhaps it would be valuable to provide some suggestion regarding how to best mitigate this potential issue.

We agree that it is not clear which algorithm is ideal or best reflective of the biology of the systems tested here. We expanded how this kind of work can be approached in the future in the Discussion.

“In addition, codon optimization algorithms are updated with new information periodically with the assumption that they yield improved results, although it is unclear in practice whether an algorithm is better optimized to produce results that reflect the true biology of a transgene. […] This approach could ensure the algorithm produces a transgenic phenotype that most closely resembles the natural phenotype.”

Finally, it is yet to be demonstrated whether wmk is involved in male killing also in natural population, and to what extent there are shared patterns and mechanisms of male killing induced by other bacterial endosymbionts such as Spiroplasma.

This is true and we note relevant text in the Discussion. We aimed to keep this emphasis in the new, streamlined Discussion by bringing it up several times.

“First, as discussed, the wmk transgenic phenotypes are likely sensitive to post-transcriptional processes. This has important implications for understanding wmk in a natural male-killing context as soon as feasible techniques are available, since heterologous expression and its reliance on codon optimization may obscure our understanding of the gene’s biology.”

“However, if codon optimization potentially changes the interpretation of transgenic findings, then phenotypes should be corroborated in natural contexts once tools such as genetic editing are available in the relevant organism. Specifically, wmk should be knocked out in native contexts once it is more technically feasible.”

“If we make the assumption that wmk is involved in male killing in nature, which requires confirmation beyond transgenic recapitulation of the phenotype, then the results here give the basis for additional hypotheses that require further testing.”